DATA RELEASE

# Genomic Resources for the North American Water Vole (*Microtus richardsoni*) and the Montane Vole (*Microtus montanus*)

Drew J. Duckett[1,*], Jack Sullivan[2], Stacy Pirro[3] and Bryan C. Carstens[1]

1 Department of Evolution, Ecology, and Organismal Biology, The Ohio State University, 1315 Kinnear Rd., Columbus, OH 43212, USA
2 Department of Biological Sciences, University of Idaho, Box 443051, Moscow, ID 83844-3051, USA
3 Iridian Genomes, Inc., 6213 Swords Way, Bethesda, MD 20817, USA

## ABSTRACT

Voles of the genus *Microtus* are important research organisms, yet genomic resources are lacking. Such resources would benefit future studies of immunology, phylogeography, cryptic diversity, and more. We sequenced and assembled nuclear genomes from two subspecies of water vole (*Microtus richardsoni*) and from the montane vole (*Microtus montanus*). The water vole genomes were sequenced with Illumina and 10× Chromium plus Illumina sequencing, resulting in assemblies with ~1600,000 and ~30,000 scaffolds, respectively. The montane vole was also assembled into ~13,000 scaffolds using Illumina sequencing. Mitochondrial genome assemblies were also performed for both species. Structural and functional annotation for the best water vole nuclear genome resulted in ~24,500 annotated genes, with 83% of these having functional annotations. Assembly quality statistics for our nuclear assemblies fall within the range of genomes previously published in the genus *Microtus*, making the water vole and montane vole genomes useful additions to currently available genomic resources.

**Subjects** Genetics and Genomics, Animal Genetics, Evolutionary Biology, Functional Genomics

**Submitted:** 22 January 2021

\* Corresponding author. E-mail: duckettdj@gmail.com

Preprint submitted at https://doi.org/10.1101/2021.04.04.438380

## DATA DESCRIPTION

### Background

The genus *Microtus* comprises 62 species of voles, distributed throughout North America, Europe, and Asia [1]. *Microtus* is believed to have undergone rapid speciation and diversification, with all speciation events occurring within the past 4 million years [2, 3]. It has been suggested that some nominal species, such as *M. pennsylvanicus*, contain cryptic diversity [4]. *Microtus* is an important model system across multiple biological disciplines, including studies of adaptation (e.g., [5]), infectious disease (e.g., [6]), parental care (e.g., [7]), and population dynamics (reviewed in [8]).

Currently, assembled genomes for four *Microtus* species have been deposited in GenBank: two European species (*M. agrestis* and *M. arvalis*) and two North American species (*M. ochrogaster;* [9], and *M. oeconomus*). The present study provides resources for two additional species: *M. richardsoni* (NCBI:txid111840) and *M. montanus* (NCBI:txid88450).

The North American water vole (*M. richardsoni*) occupies a large, disjunct distribution in the Pacific Northwest of North America, with habitat in the Cascades Mountains and the

Rocky Mountains, spanning from southern Canada into central Utah. Four subspecies are currently recognized: *M. r. arvicoloides* in the Cascades Mountains, *M. r. richardsoni* in the Canadian Rocky Mountains, *M. r. macropus* in the central Rocky Mountains and Wyoming, and *M. r. myllodontus* in Utah. *M. richardsoni* is adapted to a semiaquatic lifestyle, relying on alpine and subalpine streams for creating burrows and escaping predators [10]. Like other semiaquatic mammals (e.g., otters), adaptations to this lifestyle have probably been driven by natural selection [11–13]. Water voles are among the largest species of *Microtus*, and their frequent movement makes runways of trampled vegetation along streams [10, 14]. Unlike most other vole species, *M. richardsoni* does not appear to experience regular population boom and bust cycles, although population size in the species may correlate with levels of precipitation [15]. Despite being listed in the International Union for Conservation of Nature (IUCN) Red List [16] as being of '*Least Concern*', the species is listed as 'critically imperiled' in the Wyoming Natural Diversity Database owing to its specific habitat requirements, which can be substantially degraded by livestock grazing [17].

The montane vole (*M. montanus*) is partially sympatric with *M. richardsoni*, and can be found throughout most of the water vole's range, with the exception of the Canadian Rockies. However, *M. montanus* can be found farther south and east, including areas of California, Nevada, Colorado, Arizona, and New Mexico [18]. The species has been divided into 15 subspecies, including *M. m. canescens* in the Cascades Mountains, *M. m. nasus* in the central Rocky Mountains, and *M. m. amosus* in northern Utah. Notably, *M. montanus* does not exhibit a break in its range in the Columbia Basin; probably because it is not restricted to riparian areas like *M. richardsoni*. The species is IUCN Red-Listed as being of '*Least Concern*'. However, *M. m. arizonicus* is 'endangered' according to the New Mexico State Game Commission Regulation [18], and *M. m. ricularis* is 'of concern', owing to its small range and declining population size [19].

## Context

The rapid radiation of *Microtus* voles has hindered systematic classification, leading to multiple taxonomic revisions and conflicting phylogenetic analyses [1, 20, 21]. Consequently, both species boundaries and relationships between species are difficult to infer.

Genomic resources within *Microtus* will help resolve these questions, and resources have steadily increased in recent years. Here, we present two nuclear and one mitochondrial genome assembly for *M. richardsoni*, as well as a structural and functional annotation for one of the *M. richardsoni* genomes. The subspecific classifications and disjunct range of the species means that *M. richardsoni* has been included in multiple phylogeography studies in the Pacific Northwest [22–24]. These studies were based solely on mitochondrial DNA, and the results of analyses that investigated species limits and demographic history were limited to inferences that can be derived from a single gene tree. These genomic resources will provide a rich source of data to address these knowledge gaps and aid in future studies of adaptation.

We also present single nuclear and mitochondrial genome assemblies for *M. montanus*. Genomic resources in *M. montanus* will provide a wealth of data to assess subspecies boundaries, quantify gene flow among subspecies, and aid in conservation efforts of threatened subspecies. Genome-level comparisons are also made between the new genome assemblies and other *Microtus* genome assemblies to examine differences in assembly quality and repeat content.



## METHODS

### Sequencing and nuclear genome assembly

Frozen tissue from a single *M. r. arvicoloides* individual collected from the southern Cascades Mountain range (JMS_292; 44.016667N, −121.750000E; [22]) was sent to Hudson Alpha (Huntsville, AL, USA) for high molecular weight DNA extraction and 10× Chromium library preparation [25]. In the 10× method, each extracted DNA fragment receives a different barcode before the fragment is sheared for library preparation. After sequencing, these barcodes are used to connect sequencing reads for a more contiguous assembly. After sequencing with a single run on an Illumina HiSeqX, the resulting 150-bp paired-end reads were input into Supernova v. 2.2.1 for *de novo* genome assembly with –maxreads=all [26].

Additional tissue was obtained from a single *M. r. macropus* individual collected from the northern Rocky Mountains (JMG_88; 46.333333N, −114.633333E; [22]). DNA was extracted using a Qiagen DNeasy Blood and Tissue Kit, and the DNA was sent for library preparation and sequencing by Iridian Genomes, Inc (Bethesda, MD, USA). Then, 150-bp paired-end reads were sequenced on two runs of an Illumina HiSeqX. Genome assembly was performed using two different de Bruijn graph-based programs, SOAPdenovo v. 2.04 (SOAP, RRID:SCR_000689) and Discovar de novo v. 52488 (Discovar assembler, RRID:SCR_016755) [27, 28]. For SOAPdenovo, quality trimming was performed using fastQC v. 0.11.8 (FastQC, RRID:SCR_014583) and Trimmomatic v. 0.38 (Trimmomatic, RRID:SCR_011848) with settings ILLUMINACLIP: 2:30:10, LEADING:3, TRAILING:3, SLIDINGWINDOW:4:15, and MINLEN:36 [29, 30]. SOAPdenovo assemblies were performed with settings max_rd_len=150, avg_ins=300, reverse_seq=0, asm_flags=3, rd_len_cutoff=150, rank=1, pair_num_cutoff=3, and map_len=32. SOAPdenovo was run with *k*-mer values of 63, 89, 95, and 101 based on analysis of optimal *k*-mer values in kmerGenie v. 1.7051 [31]. Raw reads were used as input for *de novo* genome assembly with Discovar, as recommended in the program documentation.

To provide the most contiguous assembly for *M. richardsoni*, a hybrid assembly was performed using the ARCS+LINKS pipeline [32, 33]. The ARCS+LINKS pipeline uses barcoding information from the 10× Chromium reads to scaffold the contigs from a separate genome assembly. Barcoded reads from *M. r. arvicoloides* were mapped to the *M. r. macropus* Discovar assembly with bwa mem v. 0.7.17 [34] before converting the mapped reads to BAM format and sorting with SAMTools v. 1.8 (SAMTOOLS, RRID:SCR_002105) [35]. ARCS v. 1.0.5 and LINKS v. 1.8.6 were then run with settings –s 98 –c 5 –l 0 –z 500 –d 0 –r 0.05 –m 50-10000 –e 30000 and –d 4000 –k 20 –l 5 –t 2 –a 0.3 –o 0 –a 0.3 –z 500, respectively.

As part of a separate project, a single *M. montanus* individual from Utah (UMNH:Mamm:30891; 38.19381N, −111.5824E) was misidentified as *M. richardsoni*. DNA was extracted from the sample using a Qiagen DNeasy Blood and Tissue Kit before being sent to the University of California Davis Genome Center for library preparation and sequencing. Then, 150-bp paired-end sequences were collected with a single shared run on an Illumina NovaSeq. Species identity was confirmed using the Barcode of Life Database (BOLD [36]). Reads were checked and trimmed for quality with fastQC and Trimmomatic as above, before mapping reads to the mitochondrial cytochrome oxidase I (COI) sequence of *M. r. macropus* [37] using bwa mem. The resulting mapped reads were converted to BAM format, sorted, and indexed with SAMTools. PCR duplicates were identified and removed with Picard v. 2.3.0 (Picard, RRID:SCR_006525) [38]. Resulting reads were then piled with SAMTools mpileup using base and mapping quality scores of 30, consensus sequences were

generated with bcftools v. 1.3.1 (SAMtools/BCFtools, RRID:SCR_005227) [39], and consensus sequences were converted to fastq format using vcfutils with a minimum depth filter of 5 and maximum depth filter of 10,000 [35]. The resulting sequence was input into BOLD. Owing to low sequencing coverage, *de novo* genome assembly was not appropriate for *M. montanus*. To provide a preliminary genome sequence, a reference-guided genome assembly was performed with RaGOO v. 1.1 [40]. Raw reads were input into Discovar to generate an initial genome assembly. Misassembly correction was performed with RaGOO, using reads trimmed with the same settings as the *M. r. macropus* reads, and RaGOO was then used to scaffold the Discovar contigs onto the *M. r. arvicoloides* assembly; this is more closely related to *M. montanus* than the other available *Microtus* genome assemblies [3]. Since *M. montanus* has less than half the chromosomes of *M. richardsoni* ($2n$ = 22–24 in *M. montanus* versus 56 in *M. richardsoni* [41]), the possibility of structural errors in the *M. montanus* assembly was examined by calculating the percentage of reads that mapped back to the assembly using bwa mem and bamtools v. 2.2.2 [42].

The final assemblies were submitted to GenBank [43], where screening was performed to identify any contamination, and contaminated scaffolds were removed. All assemblies were evaluated with QUAST v. 5.0.1 (QUAST, RRID:SCR_001228) [44], bbmap v. 38.35 (BBmap, RRID:SCR_016965) [45], custom Python scripts [46], and BUSCO v. 5.0.0 (BUSCO, RRID:SCR_015008) using the Euarchontoglires reference set [47]. After comparing assembly statistics from the different assemblies of *M. r. macropus*, the Discovar assembly was selected as the best one because it had less fragmentation, higher N50 and lower L50 values, and a higher BUSCO score than the SOAPdenovo assemblies (Table 1). Genome sequencing of *M. r. arvicoloides* produced over 800 million reads and 47× genome sequencing coverage. The final genome assembly consisted of ~32,000 scaffolds, with an N50 of 2.3 Mb (megabase pairs), 1.3% missing data (N), and a BUSCO score of 85.8%. Supernova estimated the length of the genome assembled to be ~2.4 Gb (gigabase pairs) and the total genome size to be ~2.6 Gb. *M. r. macropus* sequencing produced over 600 million reads and 35× coverage. Genome assembly with Discovar resulted in ~1.6 million scaffolds, with an N50 of 16 Kb (kilobase pairs), 0.06% N, and a BUSCO score of 54.5%. Given the many programs available to perform *de novo* genome assembly from short reads, another program might have produced a more contiguous *M. r. macropus* assembly, but previous studies have shown that Discovar performs well compared to other programs [48, 49]. The hybrid assembly produced with the ARCS+LINKS pipeline had ~1.6 million scaffolds, an N50 of 38 Kb, 0.09% N, and a BUSCO score of 59.8%. Because the hybrid assembly was of poor quality, it was not used for further analyses, and the *M. richardsoni* subspecies assemblies were kept separate. High fragmentation of the hybrid assembly is probably associated with fragmentation of the Discovar input assembly. Published results with this hybrid pipeline often include a much higher sequencing coverage of the input contigs to produce a better starting point for the pipeline. Therefore, in the future, additional Illumina sequencing with *M. r. macropus* might substantially improve the hybrid assembly. To produce the preliminary *M. montanus* genome, 108 million reads (13× coverage) were used, resulting in ~13,000 scaffolds, an N50 of ~3.1 Mb, 8.8% N, and a BUSCO score of 82.6%. Additionally, 89.3% of reads mapped back to the *M. montanus* assembly.

## Mitochondrial genomes

The complete mitochondrial genomes of *M. r. arvicoloides* and *M. montanus* were assembled using the genomic sequencing reads. Mitochondrial genomes were assembled by

**Table 1.** Comparison of genome assembly strategies for *Microtus richardsoni macropus*.

|  | Discovar | SOAPdenovo | SOAPdenovo | SOAPdenovo | SOAPdenovo |
|---|---|---|---|---|---|
| *k*-mer (*n*) | NA | 63 | 89 | 95 | 101 |
| Length (Gb) | 2.54 | 2.72 | 2.88 | 2.89 | 3.21 |
| Scaffolds (*n*, millions) | 1.6 | 4.1 | 4.0 | 4.1 | 6.7 |
| Max Scaffold (Kb) | 264 | 186 | 146 | 174 | 139 |
| N50 (Kb) | 16.1 | 4.5 | 3.4 | 3.4 | 1.5 |
| L50 (*n*, thousands) | 35.7 | 117 | 156 | 163 | 371 |
| BUSCO (%) | 54.5 | 38.1 | 37.1 | 35.9 | 25.9 |
| N (%) | 0.06 | 1.45 | 0.99 | 0.94 | 0.90 |
| GC content (%) | 42.13 | 41.92 | 41.91 | 41.92 | 41.98 |

NA: not applicable.

both mapping reads to a reference mitochondrial genome, and by using the reference-guided assembly program Novoplasty v. 4.1 [50]. For the mapping assembly, reads were mapped to the *M. r. macropus* mitochondrial genome using the same steps as in the *M. montanus* BOLD analysis. The mitochondrial assemblies were 16,285 bp and 16,268 bp in length, with an average depth of coverage of 7886× and 6805× for *M. r. arvicoloides* and *M. montanus*, respectively. Reference-guided mitochondrial assemblies with Novoplasty used the *M. r. macropus* mitochondrial genome as the reference, along with settings Genome Range = 12,000–22,000, *K*-mer = 33, Read Length = 150, and Insert size = 400. Because the *M. r. arvicoloides* dataset contained many reads, 25% of reads were subsampled for assembly, as suggested in the program documentation. The assemblies for *M. r. arvicoloides* and *M. montanus* were 16,298 bp and 16,319 bp in length, with average depths of coverage of 5131× and 14,713×, respectively.

To compare mitochondrial assemblies between methods, the assemblies were aligned using the MUSCLE plugin in Geneious v. R9 (Geneious, RRID:SCR_010519) with eight iterations and an open gap score of –1 [51, 52]. This comparison showed that the Novoplasty assemblies contained multiple insertions compared with the mapped assemblies and the reference mitochondrial genome. These insertions were up to 13-bp long in multiple genes, including trnT, trnK, and ATP8. Comparison with other *Microtus* mitochondrial genomes (*M. ochrogaster*; NC_027945.1 and *M. fortis*; NC_015243.1) showed that these insertions were only present in the Novoplasty mitochondrial assemblies. Therefore, the mapping assemblies were used for further analyses. The mapping assemblies for both species included ambiguous bases, which were much more frequent for *M. montanus* than *M. r. arvicoloides*. These could be the result of using the mitochondrial genome of a different subspecies (for *M. r. arvicoloides*) or species (for *M. montanus*) for mapping the reads. Additionally, the presence of nuclear DNA of mitochondrial origin (NUMTs [53, 54]) might have influenced these results. If mitochondrial segments have been incorporated into the nuclear genomes and subsequent mutations have occurred, both nuclear and mitochondrial sequences could be mapped to the same mitochondrial region during assembly, resulting in the ambiguous bases observed here. NUMTs are likely to be present, as they have been documented in other species of *Microtus* [55–57]. Both mitochondrial genomes were annotated using MITOS [58]. The annotations each consisted of 22 tRNA genes, two rRNA genes, and 13 protein-coding genes.



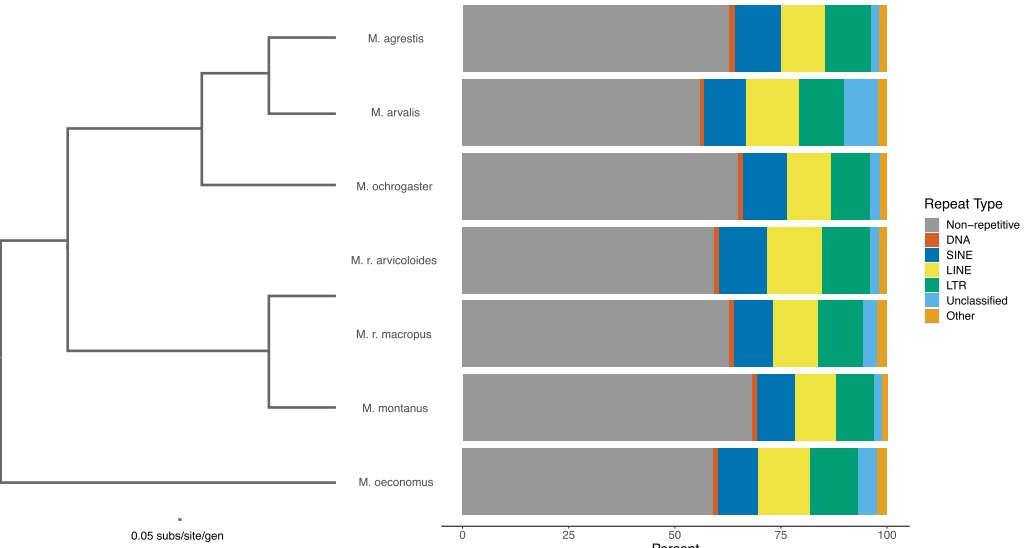

**Figure 1.** Repeat content among *Microtus* genomes. SINE: short interspersed nuclear element; LINE: long interspersed nuclear element; LTR: long terminal repeat; Other consists of small RNA, satellite, simple, and low complexity repeats. The phylogeny displayed was recreated from [3] by pruning unincluded species from the data alignment and rerunning RAxML [77] with the same settings used in the original analysis.

## *Microtus* genome assembly comparison

The available *Microtus* genome assemblies, *M. agrestis* (GCA_902806755.1), *M. arvalis* (GCA_007455615.1), *M. ochrogaster* (GCA_000317375.1), and *M. oeconomus* (GCA_007455595.1), were downloaded from GenBank. Assembly summary statistics were calculated using QUAST, bbmap, and custom Python scripts [46]. To compare repeat content among all genomes, including the three produced by the current study, repeats were first identified *de novo* using RepeatModeler v. 2.0.1 (RepeatModeler, RRID:SCR_015027) [59]. RepeatMasker v. 4.0.9 (RepeatMasker, RRID:SCR_012954) was then used to further identify repeats using a combined repeat library that included the repeats identified from RepeatModeler and those from the RepeatMasker *Rodentia* database [60]. The percentage of the genome comprising each type of repeat element was extracted from the RepeatMasker log file for each genome assembly.

All genome assemblies used some form of Illumina sequencing (Table 2), although assembly continuity varied greatly among assemblies, from 1366 scaffolds in *M. agrestis* to 1.6 million scaffolds in *M. r. macropus*. Genome coverage was similarly varied, from 13× in *M. montanus*, to 35× in *M. r. macropus*, to 77× in *M. arvalis* and *M. oeconomus*. The percentage of repetitive regions ranged from 31.7% in *M. montanus* to 44.1% in *M. arvalis* (Figure 1). Repeat content did not appear to be associated with phylogenetic relatedness, since repeats between the two subspecies of *M. richardsoni* were not more similar to each other than to other *Microtus* species. However, it is possible that the repeat content is affected by the continuity of the genome assemblies. Further research is needed to confirm this relationship.

## Genome annotation

The *M. r. arvicoloides* genome assembly was annotated with the MAKER v. 2.31.10 (MAKER, RRID:SCR_005309) pipeline [61], loosely following the tutorial by Daren Card [62]. Briefly,

**Table 2.** Genome assembly comparison among *Microtus* species.

| Species | *M. agrestis* | *M. arvalis* | *M. montanus\** | *M. ochrogaster* | *M. oeconomus* | *M. r. arvicoloides\** | *M. r. macropus\** |
|---|---|---|---|---|---|---|---|
| Distribution | Europe | Europe | North America | North America | North America | North America | North America |
| Year | 2020 | 2019 | 2020 | 2012 | 2019 | 2020 | 2020 |
| BioProject | PRJEB36805 | PRJNA551185 | PRJNA673873 | PRJNA72443 | PRJNA551187 | PRJNA673719 | PRJNA509068 |
| Sequencing | 10× Chromium + Illumina | Illumina | Illumina | Illumina | Illumina | 10× Chromium + Illumina | Illumina |
| Assembler | Supernova | Discovar | RaGOO | ALLPATHS | Discovar | Supernova | Discovar |
| Length (Gb) | 2.03 | 2.62 | 2.34 | 2.29 | 2.31 | 2.36 | 2.54 |
| Coverage (×) | 50 | 77 | 13 | 94 | 77 | 47 | 35 |
| Scaffolds ($n$) | 1366 | 1,081,432 | 12,962 | 6341 | 562,436 | 31,632 | 1,648,927 |
| Longest scaffold (Mb) | 56.96 | 0.80 | 748.72 | 126.73 | 0.93 | 16.00 | 0.26 |
| N50 (Mb) | 13.35 | 0.53 | 3.08 | 61.81 | 0.11 | 2.30 | 0.02 |
| L50 ($n$) | 45 | 11,870 | 91 | 14 | 5,556 | 278 | 35,660 |
| N (%) | 2.87 | 0.07 | 8.81 | 8 | 0.12 | 1.29 | 0.06 |
| GC (%) | 42.33 | 41.71 | 42.38 | 42.25 | 42.18 | 42.21 | 42.13 |

Assemblies with a * were produced by the present study. Note: in-depth methods for *M. agrestis* are not available, and it is possible that the assembly includes additional sequencing and/or methods.

the pipeline comprises masking repeats followed by multiple rounds of annotation, with both evidence-based and *ab initio* gene models. Repeats were identified as described above. Complex repeats were then extracted from RepeatMasker results using grep with keywords "Satellite" and "rich". Within Maker, the model_org argument was set to "simple" so Maker would soft-mask simple repeats, and the RepeatMasker results were provided to hard-mask complex repeats. Evidence-based gene discovery used protein and mRNA sequences from the previous genome annotation of *M. ochrogaster* (GCF_000317375.1), as well as an additional RNA-Seq assembly from *M. pennsylvanicus* (GSM3499528 [63]). Hidden Markov models (HMMs) for *ab initio* gene prediction were trained using both SNAP and Augustus v. 2.5.5 (Augustus: Gene Prediction, RRID:SCR_008417) [64, 65]. With SNAP, gene models identified by MAKER were filtered using an annotation edit distance (AED) of 0.5 and an amino acid length of 50. After validating these models with SNAP's Fathom utility, removing likely errors, and including 1000 bp surrounding each training sequence, the training sequences were passed to the hmm-assembler script. For Augustus, training sequences plus 1000 bp on each side were obtained from the first round of MAKER mRNA annotations. Augustus was used to train the HMM using the –long option in BUSCO and the Euarchontoglires reference set. MAKER was then run again with the previously annotated gene models, and the HMM models from SNAP and Augustus. After the initial MAKER run, two cycles of *ab initio* gene prediction and annotation with MAKER were performed. To prevent overfitting, results were compared after each round of MAKER. Because the increase in AED score was minimal between the first and second rounds of *ab initio* gene prediction, further analysis was conducted on the results after the first round only. This round annotated ~24,500 genes, with a mean gene length of 7445 bp (Table 3), which is within the range found in previous studies of *M. ochrogaster* (22,427 genes; GCF_000317375.1) and *Arvicola amphibious* (25,136 genes; GCF_903992535.1). Of these annotations, all occurred on scaffolds greater than 1 Kb in length, and 97% occurred on scaffolds greater than 10 Kb in length.

Functional annotation of the *M. r. arvicoloides* genome was performed using GOfeat [66], an online functional annotation tool that uses multiple protein databases including

**Table 3.** Structural annotation summary after each round of MAKER.

|  | Before gene modeling | Gene modeling round 1 | Gene modeling round 2 |
|---|---|---|---|
| Genes (*n*) | 20,945 | 24,548 | 23,811 |
| Exons (*n*) | 139,845 | 192,974 | 179,225 |
| mRNA (*n*) | 20,945 | 24,548 | 23,811 |
| tRNA (*n*) | - | 24,504 | 24,539 |
| 5′ UTR (*n*) | - | 1229 | 1180 |
| 3′ UTR (*n*) | - | 503 | 642 |
| Mean gene length (bp) | - | 7445 | 7132 |
| AED <0.50 (proportion) | 0.993 | 0.881 | 0.888 |
| AED <0.25 (proportion) | 0.672 | 0.543 | 0.520 |
| BUSCO (Complete) (%) | - | 67.7 | 70.5 |

UTR: untranslated region; AED: annotation edit distance. Values with dashes were not analyzed prior to gene modeling with SNAP and Augustus.

UniProt [67], InterPro[68], and Pfam [69]. An input file for GOfeat was generated by supplying the genome assembly FASTA file and the MAKER General Feature Format (GFF3) file to the Python package gffread v. 0.12.1 [70]. GOfeat annotated 83.49% of genes. Biological processes accounted for 42.46% of annotations, cellular components accounted for 30.29%, and molecular functions comprised 27.25%. The most frequent gene ontology (GO) terms were 'positive regulation of transcription by RNA polymerase II', 'negative regulation of transcription by RNA polymerase II', and 'DNA-templated regulation of transcription' for biological processes; 'cytoplasm' and 'plasma membrane' for cellular components, and 'metal ion binding' and 'calcium ion binding' for molecular functions.

## DATA VALIDATION AND QUALITY CONTROL

As described above, BUSCO was used to assess the quality of each genome assembly. Of the three final assemblies, two (*M. r. arvicoloides* and *M. montanus*) had BUSCO completeness scores over 80%. Alternatively, the low BUSCO score for the best *M. r. macropus* assembly shows that while it can still be useful for mapping reads, the assembly could be substantially improved by collecting additional data. Additionally, a high percentage of reads mapped back to the *M. montanus* genome, suggesting that the assembly is largely complete despite using the *M. r. arvicoloides* assembly as a reference.

## REUSE POTENTIAL

The current study details the assembly and annotation of three nuclear and two mitochondrial genomes. Compared with previously published nuclear genomes, the *M. r. arvicoloides* and *M. montanus* genomes are of high quality, as evidenced by the low number of scaffolds, high N50/L50 values, and high BUSCO scores. While not as complete as other *Microtus* genomes, the nuclear genome of *M. r. macropus* will still be useful for mapping low-coverage reads or reduced representation sequencing data. Furthermore, the mitochondrial genomes contributed here add to a growing number for the genus *Microtus*, and reinforce earlier suggestions that high-quality mitochondrial genomes can be obtained as byproducts of nuclear sequencing (e.g. [71, 72]). Overall, the data presented serve as an example that even though they do not include chromosomal information, high-quality draft genomes can be produced from widely available and very cost-effective methods, like the 10× Chromium protocol. These references can aid various studies, including those examining genus and species adaptation [73, 74], phylogenetics [21], phylogeography [24,

75], and disease dynamics [6, 76]. However, some activities, like exploring changes to chromosome structure, will not be possible owing to fragmentation and the lack of chromosomal mapping for these assemblies. Finally, the *M. r. macropus* and *M. montanus* sequencing data and preliminary assemblies will serve as building blocks for more accurate reference genomes in the future.

## DATA AVAILABILITY

Raw sequences, nuclear assemblies, and mitochondrial assemblies are available from GenBank under BioProjects PRJNA673719, PRJNA509068, and PRJNA673873 for *M. r. arvicoloides*, *M. r. macropus*, and *M. montanus* respectively. The custom python script used to calculate genome assembly summary information is available on GitHub [46]. Other supporting data, including assemblies, full BUSCO tables, structural annotation gff files, functional annotation tables, and repeat libraries are available in the *GigaScience* data repository GigaDB (*M. richardsoni* [78], *M. montanus* [79]).

## DECLARATIONS
## LIST OF ABBREVIATIONS

BOLD: Barcode of Life Database; bp: base pair; COI: cytochrome oxidase I; Gb: gigabase pairs; Mb: megabase pairs; Kb: kilobase pairs; AED: annotation edit distance; HMM: hidden Markov model; NUMT: nuclear DNA of mitochondrial origin.

## ETHICAL APPROVAL

Not applicable.

## CONSENT FOR PUBLICATION

Not applicable.

## COMPETING INTERESTS

SP is the director of Iridian Genomes, Inc. The other authors declare that they have no competing interests.

## FUNDING

Sequencing was funded by Iridian Genomes, Inc., as well as the National Science Foundation (DEB-1457519). Salary support for DD was provided by The Ohio State University and the National Science Foundation (DBI-1945347).

## AUTHORS' CONTRIBUTIONS

DD, JS, and BC conceived the study. JS, SP, and BC provided funding for sequencing. DD performed DNA extractions, assembled genomes, and annotated genomes with input from SP. DD and BC wrote the manuscript with input from JS and SP. DD and SP submitted the resources to GenBank. All authors approved the final version.

## ACKNOWLEDGEMENTS

We thank Jeffrey Good and Eric Rickart/Utah Museum of Natural History for tissue samples, Michael Broe for advice with genome assembly and annotation, and the Ohio Supercomputer Center (OSC) for computational resources.

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
