## [Reviewer Report]

Reviewer name and names of any other individual's who aided in reviewer Aaron ShaferDo you understand and agree to our policy of having open and named reviews, and having your review included with the published papers. (If no, please inform the editor that you cannot review this manuscript.)YesIs the language of sufficient quality?YesPlease add additional comments on language quality to clarify if needed
Are all data available and do they match the descriptions in the paper? YesAdditional CommentsAre the data and metadata consistent with relevant minimum information or reporting standards? See GigaDB checklists for examples <a href="http://gigadb.org/site/guide" target="_blank">http://gigadb.org/site/guide</a>YesAdditional CommentsIs the data acquisition clear, complete and methodologically sound?YesAdditional CommentsIs there sufficient detail in the methods and data-processing steps to allow reproduction?NoAdditional CommentsI would include all flags for assemblies even if default; unclear how the 10x + Illumina data were integrated (if at all) - see comments belowlIs there sufficient data validation and statistical analyses of data quality? YesAdditional CommentsI suppost BUSCO and gene number is a form of validation. Is the validation suitable for this type of data?YesAdditional CommentsIs there sufficient information for others to reuse this dataset or integrate it with other data?NoAdditional CommentsSee comment below; while the short-read data is great, the genomic resource I likely would reassemble for a variety of reasons outlined in Additional Comments.Any Additional Overall Comments to the AuthorThe paper is well written, and I have no comments about the the content - well done here. My main concern lies with the genome resources - and in this case I would likely use the raw data, rather than the assemblies provided. I offer my rationale and suggestions:

My lab was heavily pushed by a colleague towards the use of Meraculous in our short-read assembly of mammal genomes ( https://jgi.doe.gov/data-and-tools/meraculous/ ) ; this is because it’s really designed for short-read assemblies of big genomes (i.e. no addition of mate-pair) AND it performs very well in the Assemblathon metrics https://academic.oup.com/gigascience/article/2/1/2047-217X-2-10/2656129 - notably Figure 16-18 you start to see clear differences between meraculous and say soapdenovo. Thus for just the Illumina data I would very much like to see a more appropriate assembly explored as stats like N50 and no. scaffolds will likely improve considerably with the appropriate methods.

Likewise, it’s very unclear in the methods how M. r. arvicoloides was assembled: I see SUPERNOVA for the 10X data (great), and probably soapdenovo for the Illumina data (see above). But how were they combined? This sequencing strategy is really designed for a hybrid assembly (see for example DGB2OlC https://github.com/yechengxi/DBG2OLC) this is appropriate for 10X data and really does work! But there are others. 

Note M. agretus that has an identical sequencing strategy to M. r. arvicoloides almost has ~3% the total scaffolds – follow whatever they did! And I will say, while the authors state their genome is on par with other Microtus, this appears true by Table 3, only M. agretus currently has an assembly that I think is at current standards. The level of fragmentation and low BUSCO scores really support re-visiting the assembly suggestions, as I think the current .fasta will be of limited utility in a population or comparative genomics study. 

The gene number is pretty high for a mammal and I worry that’s due to fragmentation. It would be reasonably to only annotate scaffolds >10Kb or 50KB, but then there’s not much of a genome left. Ideally the bulk of your genome (>>90%) would fall on these scaffolds. There is really no sense annotation your small fragments (have you tested for contamination? Note NCBI will do this before allowing for it to be deposited so I suggest it). 

You also align your data to mt genome, this is different than assembling it. You could assemble it (e.g. https://bmcbioinformatics.biomedcentral.com/articles/10.1186/s12859-017-1927-y) and that might be interesting to see if there any differences

I wish I could be more positive; an assembly like Mercaculous would take a week or so, and so would the hybrid approach, but would be worth it based on my experience with these data.
RecommendationMajor Revision

---

## [Reviewer Report]

Reviewer name and names of any other individual's who aided in reviewer Joana DamasDo you understand and agree to our policy of having open and named reviews, and having your review included with the published papers. (If no, please inform the editor that you cannot review this manuscript.)YesIs the language of sufficient quality?YesPlease add additional comments on language quality to clarify if needed
Are all data available and do they match the descriptions in the paper? YesAdditional CommentsAre the data and metadata consistent with relevant minimum information or reporting standards? See GigaDB checklists for examples <a href="http://gigadb.org/site/guide" target="_blank">http://gigadb.org/site/guide</a>YesAdditional CommentsIs the data acquisition clear, complete and methodologically sound?YesAdditional CommentsSee additional overall comments.Is there sufficient detail in the methods and data-processing steps to allow reproduction?YesAdditional CommentsIs there sufficient data validation and statistical analyses of data quality? YesAdditional CommentsSee additional overall comments.Is the validation suitable for this type of data?YesAdditional CommentsIs there sufficient information for others to reuse this dataset or integrate it with other data?YesAdditional CommentsAny Additional Overall Comments to the AuthorThe genomes presented in these work will be extremely valuable tool for Microtus related research. The manuscript is very clear and easy to follow. I have, however, a couple of comments that I hope will further improve it.
(1) Line 123: I believe more details on the measures used for the selection of the best M. r. macropus are needed. Even though the contiguity of the Discovar genome assembly is higher than the ones generated with SOAPdenovo, the BUSCO score is relatively low (54.5% versus 84% in M. r. arvicoloides, e.g.). Were the BUSCO scores for the other assemblies even lower? Is the Discovar assembly size closer to the estimated genome size?
(2) Line 131/251: Was there any genome structure verification step for the M. montanus genome assembly? For instance, which percentage of the Illumina reads could be mapped back to the finished genome assembly? 
(3) Line 131/251: Was there a reason not to use a published reference-guided assembly method (e.g. RaGOO and those listed therein) for the assembly of M. montanus genome? These could maybe further improve the assembly or help identify misassemblies.
(4) Line 180: the high difference between BUSCO scores for each M. richardsoni subspecies makes me believe that the completeness of the genomes is quite different and the fraction of the genome within repeats might be underrepresented in M. r. macropus and that the subspecies values might be closer than noted here. It is, however, difficult to depict phylogenetic relatedness from Fig. 1 for the other species, for non-experts as myself. It would be helpful to have a phylogeny next to the graph showing species relationships.
(5) Please verify Tables 1 and 2. The statistics presented for M. r. macropus do not match for N50 and longest scaffold size.RecommendationMinor Revision

---

## [Reviewer Report]

Reviewer name and names of any other individual's who aided in reviewer Bettina HarrDo you understand and agree to our policy of having open and named reviews, and having your review included with the published papers. (If no, please inform the editor that you cannot review this manuscript.)YesIs the language of sufficient quality?YesPlease add additional comments on language quality to clarify if needed
Are all data available and do they match the descriptions in the paper? YesAdditional CommentsAre the data and metadata consistent with relevant minimum information or reporting standards? See GigaDB checklists for examples <a href="http://gigadb.org/site/guide" target="_blank">http://gigadb.org/site/guide</a>YesAdditional CommentsIs the data acquisition clear, complete and methodologically sound?YesAdditional CommentsIs there sufficient detail in the methods and data-processing steps to allow reproduction?YesAdditional CommentsIs there sufficient data validation and statistical analyses of data quality? YesAdditional CommentsIs the validation suitable for this type of data?YesAdditional CommentsIs there sufficient information for others to reuse this dataset or integrate it with other data?YesAdditional CommentsAny Additional Overall Comments to the AuthorRecommendationMinor Revision